# Parenterally Administered P24-VP8* Nanoparticle Vaccine Conferred Strong Protection against Rotavirus Diarrhea and Virus Shedding in Gnotobiotic Pigs

**DOI:** 10.3390/vaccines7040177

**Published:** 2019-11-06

**Authors:** Ashwin Ramesh, Jiangdi Mao, Shaohua Lei, Erica Twitchell, Ashton Shiraz, Xi Jiang, Ming Tan, Lijuan Yuan

**Affiliations:** 1Department of Biomedical Sciences and Pathobiology, Virginia-Maryland College of Veterinary Medicine, Virginia Tech, Blacksburg, VA 24060, USA; akramesh@vt.edu (A.R.); jmao2018@vt.edu (J.M.); lsh2013@vt.edu (S.L.); etwitche@vt.edu (E.T.); aks2026@vt.edu (A.S.); 2Division of Infectious Diseases, Cincinnati Children’s Hospital Medical Center, Cincinnati, OH 45229, USA; Jason.Jiang@cchmc.org (X.J.); Ming.Tan@cchmc.org (M.T.); 3Department of Pediatrics, University of Cincinnati College of Medicine, Cincinnati, OH 45229, USA

**Keywords:** rotavirus nanoparticle vaccine, gnotobiotic pigs

## Abstract

Current live rotavirus vaccines are costly with increased risk of intussusception due to vaccine replication in the gut of vaccinated children. New vaccines with improved safety and cost-effectiveness are needed. In this study, we assessed the immunogenicity and protective efficacy of a novel P24-VP8* nanoparticle vaccine using the gnotobiotic (Gn) pig model of human rotavirus infection and disease. Three doses of P24-VP8* (200 μg/dose) intramuscular vaccine with Al(OH)_3_ adjuvant (600 μg) conferred significant protection against infection and diarrhea after challenge with virulent Wa strain rotavirus. This was indicated by the significant reduction in the mean duration of diarrhea, virus shedding in feces, and significantly lower fecal cumulative consistency scores in post-challenge day (PCD) 1–7 among vaccinated pigs compared to the mock immunized controls. The P24-VP8* vaccine was highly immunogenic in Gn pigs. It induced strong VP8*-specific serum IgG and Wa-specific virus-neutralizing antibody responses from post-inoculation day 21 to PCD 7, but did not induce serum or intestinal IgA antibody responses or a strong effector T cell response, which are consistent with the immunization route, the adjuvant used, and the nature of the non-replicating vaccine. The findings are highly translatable and thus will facilitate clinical trials of the P24-VP8* nanoparticle vaccine.

## 1. Introduction

Human rotavirus (HRV) is a leading cause of severe, dehydrating gastroenteritis in children under five years of age. Although two live-attenuated oral vaccines, RotaTeq^®^ and Rotarix^®^, have been implemented as part of national vaccination programs in over 98 different countries [1], their vaccine efficacy was reported to be lower in low- and middle- income countries (LMICs [39%–70%]) compared to that in high-income countries (80%–90%) [2,3,4,5,6,7]. A combination of factors have been suggested to be responsible for the lower efficacy, which have been reviewed and discussed in detail by Arnold [8] and Desselberger [9]. Both these live vaccines have also been linked to a low risk of intussusception (1–7 cases per 100,000 vaccinated infants) as a result of the vaccine rotavirus (RV) strains replicating in the intestines [10,11], and have remained expensive even after Gavi, the Vaccine Alliance, subsidization [12], particularly in resource-deprived countries. These scenarios have increased the demand for a safer, more cost-effective, and more efficacious vaccine, especially in LMICs that can be easily administered. 

Parenteral intramuscular (IM) vaccines have been preferred to oral vaccines due to their increased immunogenicity. They are also not directly affected by microbiome composition or gut enteropathy, both of which have been known to affect the efficacy of oral vaccines in LMICs [13]. Non-replicating rotavirus vaccines (NRRV) have been proposed as safer alternatives to live-attenuated vaccines as they do not lead to intussusception due to their parenteral immunization route [14].

The VP4 region of RV constitutes surface spike proteins that are cleaved by host intestinal proteases into two fragments, VP5* and VP8*. VP8* forms the distal portion of the VP4 spikes, interacting with glycan receptors to facilitate viral attachment and entry [15,16]. VP8* expressed in various culture systems has been explored as an immunogen in rotavirus vaccine development [17,18,19,20,21,22]. VP8*-containing vaccine candidates have been shown to induce rotavirus-specific neutralizing antibodies and/or protection in mouse, guinea pigs, and gnotobiotic (Gn) pig models [23,24,25,26,27,28]. Among those, the leading candidate is the P2-VP8 (VP8* fused to a universal tetanus toxin CD4+ T cell epitope P2) vaccine, adsorbed with aluminum hydroxide for IM administration, which has progressed to phase 1/2 human clinical trials [19,29]. 

The norovirus (NoV) P particle, referred to as the P24 particle, is an octahedral nanoparticle (≈840 kDa) composed of 24 copies of the protrusion (P) domain of the NoV capsid protein. It can be easily produced in large quantities using an *E. coli* expression system. The distal surface of each P domain, corresponding to the outermost surface of the P particle, contains three surface loops, which can tolerate large sequence insertions. Based on this concept, a nanoparticle vaccine was developed by inserting the HRV VP8* antigen into the loop sections of the P domains. The P24-VP8* nanoparticle consists of a 24-valent core of NoV P particle and 24 surface-displayed HRV VP8*s. The P24-VP8* nanoparticle shares the features of the P24 particle in self-formation, easy production, and high stability over a wide range of temperatures [30]. Efficacy studies in mice revealed that the P24-VP8* nanoparticle vaccine is highly immunogenic and capable of inducing a significantly higher VP8* specific antibody response as compared with free VP8* particles even without adjuvant [30]. 

The main objectives of this study were to assess the immunogenicity and protective efficacy of a novel P24-VP8* nanoparticle vaccine using the gnotobiotic (Gn) pig model of human rotavirus infection and disease. The Gn pig model of HRV (Wa, G1P [8]) infection and diarrhea has been well established and used in the pre-clinical evaluation of HRV vaccine efficacies [31]. No other conventional lab animals develop diarrhea after HRV inoculation [32]. Pigs are genetically, physiologically, anatomically, and immunologically similar to humans [33,34,35], allowing data from Gn pigs to be translated to humans. The immunogenicity and protective efficacy of the P24-VP8* nanoparticle vaccine were determined using the Gn pig model of HRV infection and disease. High serum IgA, IgG, and virus-neutralizing (VN) antibody titers, as well as HRV-specific IFN-γ producing T cells, have been correlated with protection from HRV infection and disease, and data has been demonstrated to be comparable in Gn pigs and human studies [24,36,37].

## 2. Materials and Methods

### 2.1. Human Rotavirus

The virulent HRV (VirHRV) inoculum consisted of a pool composed of intestinal contents collected from the 27th passage in Gn pigs of the Wa strain HRV based on successive passages carried out in Gn pigs. A total of 1 × 10^5^ fluorescent focus-forming units (FFUs) of VirHRV were diluted in 5 mL of Diluent #5 [minimal essential media (MEM, ThermoFisher Scientific); 100 IU of penicillin per mL, 0.1 mg of dihydrostreptomycin per ml; and 1% HEPES] for the inoculation of each Gn pig. The median infectious dose (ID_50_) and median diarrhea dose (DD_50_) of the VirHRV in Gn pigs were determined as approximately 1 FFU [38].

The cell culture-adapted HRV Wa strain (AttHRV), derived from the 35th passage in African green monkey kidney cells (MA104, ATCC# CRL-2378.1) [38,39], were used as the positive control for the assessment of RV antigens in feces using enzyme-linked immunosorbent assay (ELISA). The origination and passage history of the VirHRV and AttHRV have been explained by Wentzel et al. [40].

### 2.2. Vaccine

The P24-VP8* vaccine was comprised of 200 μg of P24-VP8* proteins and 600 μg Aluminum hydrogel (Al(OH)_3_) adjuvant. The vaccine was stored at 4 °C (up to 8 months) until administered to Gn pigs (Appendix A). The dosage of the P24-VP8* vaccine was selected based on a similar VP8* molar amount of the P2-VP8 subunit vaccine used in the clinical trial [19,27,29]. The VP8* region used in this vaccine was designed based on the sequence of Wa HRV. As the negative control, the Al(OH)_3_ adjuvant (G-Biosciences, St. Louis, MO, USA) was diluted in sterile PBS to form a final concentration of 600 μg/mL and stored at room temperature, as per manufacturer instructions, until administered.

The purified P24-VP8* proteins were used as the detector antigen in ELISA for the detection of serum IgA and IgG antibody responses [41] and as stimulating antigen in the intracellular IFN-γ staining assay [39,42].

### 2.3. Gn Pigs and Treatments

Pigs (Large white cross breed) used in this study were surgically derived by hysterectomy and maintained in sterile isolators, as described previously [43]. The sterility status of the pigs housed in the gnotobiotic isolators was confirmed by culturing isolator swabs and pig rectal swabs on blood agar plates and in thioglycolate broth first at 3 days after derivation and then repeated once a week until the end of the study. Pigs were fed on a diet that solely consisted of commercial UHT sterile whole cow’s milk (The Hershey Company, Hershey, PA, USA) until post-inoculation day (PID) 21, and were switched over to Similac^®^ baby formula (Abbott Laboratories, Chicago, IL, USA) until the end of the study.

A total of 25 pigs were assigned to two groups, and a subset of pigs in each group were euthanized either pre-challenge (PID 28) or at post-challenge day (PCD) 7 (Table 1).

Pigs were administered IM with an equal volume (1 mL) of either P24-VP8* vaccine formulated with adjuvant or adjuvant alone at 5 days of age (PID 0), followed by two booster doses at PID10 and PID21. The Phase I and Phase II clinical trials carried out to evaluate the effects of P2-VP8* vaccine demonstrated that participants who received a 3-dose vaccination regime shed fewer attenuated rotavirus in feces as compared to trial participants who received two doses [19,29]. Based on this rationale, we opted to use the 3-dose regimen in this current study. The timing of 3 injections in Gn pigs are established in previous studies [27,37] based on the time needed to prime and boost immune responses in Gn pigs. Serum was collected at PID 0, PID 10, PID 21, PID 28, and PCD 7 for the detection of VP8*-specific IgA, IgG, and Wa HRV-specific neutralizing antibody responses.

One subset of pigs (*n* = 3–7) from each group was euthanized before the challenge at PID 28. Another subset of pigs (*n* = 7–8) was orally challenged with 1 × 10^5^ FFU of VirHRV Wa strain and monitored from PCD 0 to PCD 7 to assess the protection against virus shedding and diarrhea conferred by the vaccine before euthanasia on PCD 7. The pathogenesis of the Wa VirHRV infection has been studied in detail in Gn pigs; diarrhea and virus shedding persisted between 4 to 7 days post infection [33,38,44,45]. Based on these observations, we limited the study duration to 7 days post-challenge in order to assess the immediate protection conferred by the vaccine against VirWa challenge. Four milliliters of 200 mM NaHCO_3_ were given orally 15–20 min before the VirHRV challenge to reduce stomach acidity to allow for rotavirus inoculum to pass through the stomach without being degraded due to low pH in the stomach.

At euthanasia, small and large intestinal contents (SIC and LIC) were collected from all pigs and processed, as described [46], for the detection of intestinal antibody responses by ELISA. Ileum, blood, and spleen were collected, and mononuclear cells (MNCs) were isolated from them for the detection of effector T cell responses by flow cytometry as described [42].

### 2.4. Assessment of Diarrhea and Detection of RV Shedding in Feces by Rotavirus Antigen ELISA and CCIF

The pigs were on a milk-based diet throughout the duration of the study, making their fecal consistency resemble that of a newborn infant. For the assessment of diarrhea, fecal consistency was recorded daily from PCD 0–7 and categorized as follows: 0: normal; 1: pasty; 2: semi-liquid; 3: liquid. The fecal scoring system used here has been well established and used for multiple Gn pigs studies [32,33,38,44,45,47,48]. Pigs were considered to be having diarrhea when their daily fecal consistency scores were recorded to be 2 or greater (≥2).

Rectal swabs were collected daily to monitor virus shedding by ELISA (for the detection of RV antigens) and cell culture immunofluorescence (CCIF; for the detection of infectious virions) from PCD 0–7. Rectal swabs were processed, as reported previously [49]. ELISA and CCIF assays for the detection and titration of VirHRV antigen in rectal swabs were carried out as previously described [33,38,44,45,47,50,51,52]. CCIF titers [fluorescent focus units (FFU)/mL)] were determined by Equation (1):(1)$$\frac{ffu}{mL}=\frac{\#\;Plaques\;counted}{d\;x\;V}$$
where d = dilution factor, and *V* = volume of virus added.

### 2.5. RV-Specific Serum VN, and VP8*-Specific Serum and Intestinal IgA and IgG Antibody Titration

VN antibody titers in serum samples were determined based on methods described previously [47]. MA104 cells were cultured in 96-well plates until an even monolayer was formed (≈3–4 days). Cells were washed once with sf-EMEM, and enriched with 100 µL of sf-EMEM and incubated at 37 °C for 2 h. The media was then discarded, and the cells were inoculated with trypsin-activated AttHRV (4 × 10^3^ FFU in 10 μg/mL trypsin) in the absence or presence of 4-fold decreasing concentrations of Gn pig serum samples. The inoculum was discarded, and the plates were incubated at 37 °C for 18 h in 5% CO_2_ containing fresh sf-EMEM. The remainder of the steps has been described in detail in a previous publication [47]. The VN antibody titer was expressed as the reciprocal of the serum dilution, which reduced the number of fluorescent cell-forming units by >80% compared to the negative control serum. VP8*-specific IgA and IgG antibody titers in serum and intestinal contents were measured by using isotype-specific antibody ELISA with purified P24-VP8* as detector antigen at the plate coating concentration of 6.63 µg/mL, following methods described elsewhere [46,47,53]. When loading the testing samples on ELISA plates, four-fold serial dilutions of each sample started from 1:4 to 1:16384 for IgA, SIC, and LIC and 1:256 to 1:1048576 for IgG.

### 2.6. Flow Cytometry

Mononuclear cells (MNCs) collected from the ileum, blood, and spleen were diluted to a concentration of 2 × 10^6^ cells/mL and were seeded into 12-well plates and stimulated with 12 μg/mL of P24-VP8* antigen for 17 h at 37 °C in 5% CO_2_ as determined previously [42]. CD3+CD4+ and CD3+CD8+ cell surface marker staining and IFN-γ intracellular staining have been described in previous publications [42,47,54,55]. All samples were stored in 0.05 mL of stain buffer and were maintained at 4 °C. A minimum of 100,000 events were acquired using a FACSAria flow cytometer (BD Biosciences, San Jose, CA, USA). Flow cytometry data were analyzed using FlowJo X software (Tree Star, Ashland, OR, USA).

### 2.7. Statistical Analysis

Gn pigs were randomly assigned into treatment groups upon derivation regardless of gender and body weight by animal care technicians. Student’s *t*-test was used for comparisons of virus shedding and diarrhea data between the treatment groups. One-way analysis of variance (ANOVA) (General linear model) was used to compare rotavirus-specific IgA, IgG, virus-neutralizing (VN) antibody titers between the treatment groups. Tukey-Kramer HSD was used for the comparison of different time points within the same treatment group. Two-way ANOVA, followed by a Multiple *t*-test, was used for comparisons of frequencies of IFN-γ producing T cells between treatment groups. ANOVA analyses were carried out using JMP 14.0 (SAS Institute, Kerry, NC, USA), and all other statistical analyses were performed using GraphPad Prism 7.0 (GraphPad Software, San Diego, CA, USA). A *p* value lower than 0.05 was accepted to be statistically significant.

## 3. Results

### 3.1. Protection against Diarrhea and Virus Shedding upon Challenge with VirHRV

Vaccinated and control Gn pigs were challenged with VirHRV at PID 28 and were monitored daily for clinical signs (diarrhea) and virus shedding from PCD 1 to PCD 7. Gn pigs that were administered with P24-VP8* vaccine had a significantly delayed onset of diarrhea (from 1.6 to 4.4 days), a significantly reduced duration of diarrhea (from 6.0 to 3.3 days), significantly lower mean diarrhea scores on PID 1 and 2, and a significantly lower cumulative fecal consistency score (from 14.3 to 9.1) as compared to the mock-vaccinated control group (Table 2 and Figure 1A). A delayed onset of virus shedding, a reduced peak titer, a reduced cumulative virus titer (presented as the area under the curve, AUC), and a significantly reduced duration (from 5.9 to 2.5 days) of virus shedding were observed in P24-VP8* vaccinated pigs when compared to the control group (Table 2). In addition, the mean daily virus shedding titer in the vaccinated pigs was significantly reduced at PCD 2 (Figure 1B), and the reduction of total virus shed (AUC) was 2.27-fold compared to the control pigs (Table 2). However, the vaccine did not significantly reduce the incidence (%) of diarrhea and virus shedding (Table 2).

### 3.2. Strong VP8*-Specific IgG and Virus Neutralizing, but Lack of IgA, Antibody Responses in Serum

In order to monitor the development of VP8* specific humoral immunity, serum samples were collected during the time of vaccine administration (PID 0, PID 10, and PID 21) at the VirHRV challenge (PID 28) and upon euthanasia (PCD 7). Serum IgG and IgA antibody responses were quantified using ELISA and depicted in Figure 2A and Figure 2B, respectively. P24-VP8*-specific IgG antibody titers in serum were significantly higher (*p* < 0.001) in vaccinated pigs at PID 10, PID 21, PID 28, and PCD 7 when compared to pigs in the control group (Figure 2A). However, serum IgA titers were only detectable after challenge (PCD 7) with VirHRV (Figure 2B).

HRV neutralizing antibodies were detected in the serum of P24-VP8* vaccinated pigs starting from PID 21 and were observed to increase similarly with VP8*-specific IgG titers until euthanasia at PCD 7. In control pigs, VN antibodies were only detectable after challenge with VirHRV and were at significantly lower levels compared to the vaccinated pigs (Figure 2C).

### 3.3. Lack of P24-VP8* Specific Antibody Responses in the Intestines

P24-VP8*-specific IgA and IgG antibody titers in SIC and LIC, collected at the time of euthanasia (PID 28 and PCD 7), were measured by ELISA. The P24-VP8* vaccine did not induce any detectable intestinal IgA or IgG antibody responses before the challenge at PID 28. After the challenge, among the eight vaccinated and challenged pigs, only VP8*-specific IgG antibodies were detected (ELISA titers ranging from 256 to 1024) in the SIC of three pigs at PCD 7 (Appendix A). However, the SIC IgG titers were not associated with the severity of diarrhea or the amount of virus shed in the three pigs throughout the challenge period.

### 3.4. P24-VP8* Vaccine did not Induce Strong VP8*-Specific Effector T Cell Responses in Intestinal and Systemic Lymphoid Tissues

Frequencies of IFN-γ+CD8+ and IFN-γ+CD4+ T cells in ileum, peripheral blood (PBL), and spleen at PID 28, and PCD 7 are summarized in Figure 3. At PID 28, slightly higher (not statistically significant) IFN-γ producing CD4+ and CD8+ T cell responses to the P24-VP8* antigen was detected in vaccinated pigs as compared to control pigs (Figure 3A). P24-VP8* vaccinated pigs had higher frequencies of IFN-γ+CD4+ T cells in ileum and blood and higher IFN-γ+CD8+ T cells in ileum, blood, and spleen compared to the mock-vaccinated control pigs. Upon the VirHRV challenge, there was still no significant difference in the frequencies of IFN-γ secreting CD4+ and CD8+ T cells between the two groups in the intestinal (ileum) or the systemic tissues (PBL and spleen) (Figure 3B).

## 4. Discussion

In this study, the immunogenicity (antibody and T cell responses) and protective efficacy of the P24-VP8* nanoparticle vaccine were evaluated in Gn pigs. We first demonstrated that the IM P24-VP8* vaccine conferred significant protection against infection and diarrhea when challenged with the homotypic virulent strain Wa of HRV. This was indicated by the significant reduction in the mean duration of diarrhea, virus shedding in feces, and significantly lower fecal cumulative consistency scores recorded from PCD 1–7 in vaccinated pigs compared to the controls. However, the vaccine did not significantly reduce the incidence (%) of diarrhea and virus shedding, indicating that there was a lack of protective immune effectors at the site of infection (small intestine) at the time of challenge, which is consistent with the observed intestinal immune responses. The IM P24-VP8* vaccine with Al(OH)_3_ adjuvant was highly immunogenic in Gn pigs. It induced strong VP8*-specific serum IgG and virus-neutralizing antibody responses from PID 21 to PCD 7 but did not induce serum or intestinal IgA antibody responses or a strong effector T cell response. These results are consistent with the IM immunization route, the Al(OH)_3_ adjuvant, and the nature of the non-replicating vaccine. Non-replicating vaccines are typically ineffective in inducing effector T cell responses. The Al(OH)_3_ adjuvant is characteristic for its ability to enhance a Th2 type immune response, promoting strong humoral responses and suppressing effector T cell responses [56].

The observed protection and immune responses data together suggest that the protection conferred by the P24-VP8* vaccine against diarrhea and virus shedding upon challenge with the virulent Wa HRV was mediated by the vaccine-induced antibodies in the serum. Although there were no antibodies present at the lumen of the small intestine, the site of HRV infection, at the time of challenge to totally prevent the initiation of RV infection, the viruses disseminated into blood from the infected small intestinal epithelial cells could have been neutralized by the high titers of VP8*-specific IgG and virus-neutralizing antibodies during the phase of viremia. Such mechanisms can reduce the chance of infection of more epithelial cells by the virus from the basolateral side [45]. Studies showing that passively transferred serum antibodies can suppress or delay viral infection in RV-challenged pigtailed macaques [57], and an inactivated IM HRV vaccine (CDC-9) reduced virus shedding in Gn pigs upon challenge with Wa VirHRV [48] likely share the same protection mechanism with the P24-VP8* vaccine. The serum IgG and virus-neutralizing antibody responses induced by the P24-VP8* IM nanoparticle vaccine had similar dynamics and magnitude as the aluminum phosphate adjuvanted inactivated CDC-9 and P2-VP8* IM vaccines in Gn pigs [27,48]. The P24-VP8* vaccine demonstrated a similar degree of protection against diarrhea but a stronger protection against virus shedding in Gn pigs as compared to the P2-VP8* vaccine [27].

There was a trend of inverse correlation between serum VP8*-specific IgG titers at PID 28 and cumulative diarrhea scores post-challenge in the vaccinated pigs (Pearson’s rank correlation, r = −0.6699 and *p* = 0.0691), suggesting that vaccinated pigs with higher serum VP8*-specific IgG responses are more likely to be protected against severe diarrhea, which is in agreement with the study of serum IgG antibody in human adults showing that VP4-specific IgG titer was correlated with resistance to HRV infection [58]. The presence of high preexisting IgG titers was also correlated with less severe or shorter duration of diarrhea among children under three years of age [59]. As reviewed by Jiang et al., serum antibodies, if present at critical levels, are either protective themselves or are an important and powerful correlate of protection against rotavirus disease [60].

Additional investigations are required to explore the full potential of P24-VP8* vaccine efficacy. First, P24-VP8* is a candidate dual-vaccine against both NoV and RV, but we only examined the immune responses and protection against HRV, not human norovirus (HuNoV). Further studies in the Gn pig model of HuNoV infection are needed to evaluate its efficacy against NoV. Second, we only examined the protection against challenge with a homotypic HRV, and it remains to be determined whether the P24-VP8* vaccine would be effective in protecting against heterotypic HRV, as the monovalent P[8] HRV vaccine Rotarix showed significant efficacy against P[4] (70.9%) and P[6] (55.2%) HRV associated gastroenteritis in African infants [61]. One of the important potential advantages of the novel P24-VP8* nanoparticle dual vaccine is that the vaccine can be formulated as a cocktail vaccine to cover multiple types of RVs and NoVs for broad protection. Thus far, the Gn pig model of HRV infection and diarrhea has only been evaluated using the P [8] Wa HRV, requiring the need to test the effectiveness of Gn pigs as a suitable model for additional HRV P types to evaluate the broadness of protection of the novel P24-VP8* nanoparticle vaccine.

## 5. Conclusions

The P24-VP8* vaccine candidate is a typical nanoparticle vaccine with 24 copies of the major RV surface neutralizing antigen VP8* displayed on the self-assembled norovirus P24 particles. The P24-VP8* nanoparticles are easily produced in *E. coli* with a high yield and simple purification procedures at a low cost. Significant enhancement of the immunogenicity of both VP8* and P domain backbone have been demonstrated in mouse immunization studies. In this current study, the usefulness of the P24-VP8* vaccine was assessed in a Gn pig model, followed by the challenge of HRV. Three doses of IM immunization of Gn pigs demonstrated the nanoparticle vaccine’s effectiveness to significantly shorten the duration of HRV diarrhea and virus shedding, reduce the severity of diarrhea, and lower the amount of virus shed when challenged. Immune responses associated with protection include high titers of VP8*-specific serum IgG antibodies and virus-neutralizing antibodies induced by the vaccine after the second and third booster doses. These findings will facilitate clinical trials of this vaccine candidate into a useful, safe, non-replicating, parental vaccine against RVs.

## Figures and Tables

**Figure 1 vaccines-07-00177-f001:**
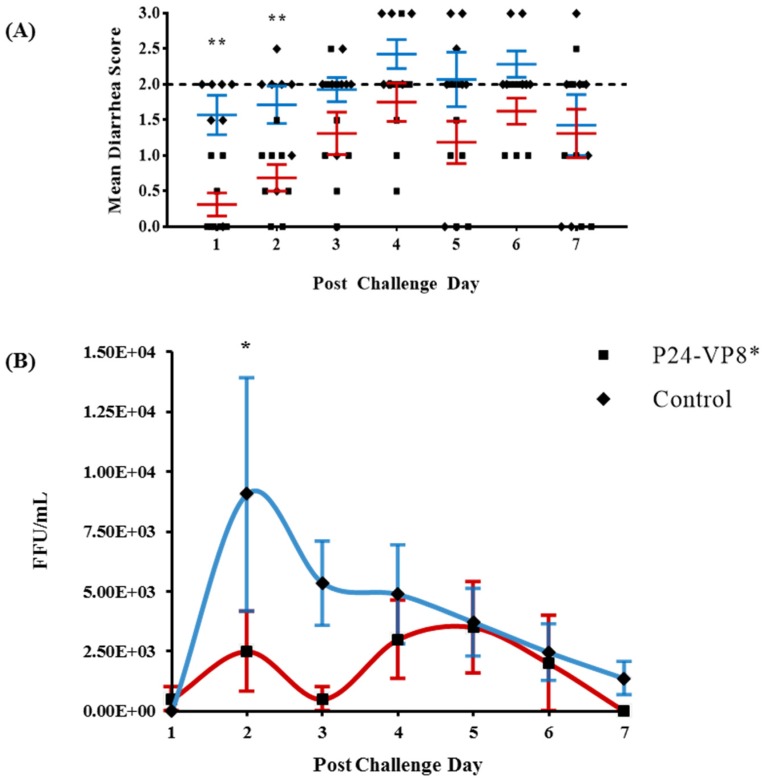
P24-VP8* vaccine protected against VirHRV diarrhea and reduced overall virus shed among vaccinated pigs. Fecal consistency (**A**) and virus shedding (**B**) were monitored daily from post challenge day (PCD) 1 to PCD 7 after the challenge with VirHRV. Fecal consistency scores ≥2 were considered to be diarrheic (dashed line indicates the threshold of diarrhea). Statistical significance between vaccinated and control groups, determined by multiple t tests, are indicated by asterisks (*, *p* < 0.05; **, *p* < 0.01).

**Figure 2 vaccines-07-00177-f002:**
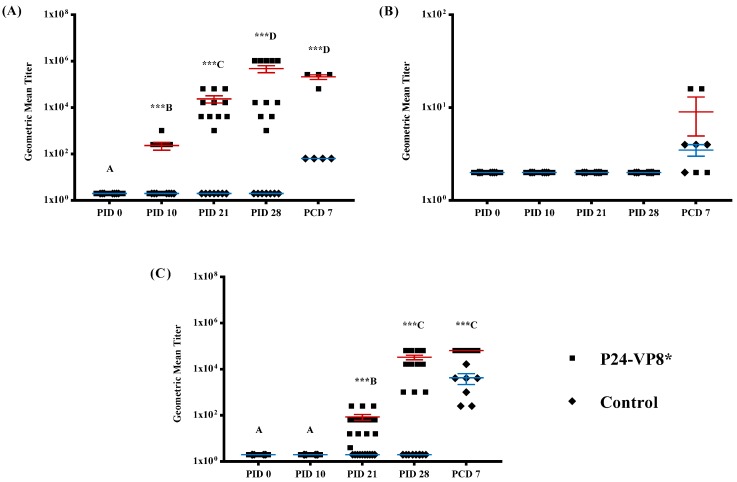
Geometric mean VP8*-specific IgG (**A**) and IgA (**B**) and Wa-HRV neutralizing (**C**) antibody titers in serum collected from Gn pigs at PID 0, 10, 21, 28, and PCD 7. Pigs were vaccinated with P24-VP8* vaccine or Al(OH)_3_ adjuvant only. Each serum specimen was tested at an initial dilution of 1:4. Negative samples were assigned an arbitrary value of 2 for calculation and graphical illustration purposes. Comparisons between groups at the same time points were carried out using Student’s *t*-test and significant differences are identified by *** (*n* = 10–15; *p* < 0.001). Tukey-Kramer HSD was used for the comparison of different time points within the same group, where different capital letters (A, B, C, D) indicate a significant difference, *p* < 0.01, and shared letters indicate no significant difference.

**Figure 3 vaccines-07-00177-f003:**
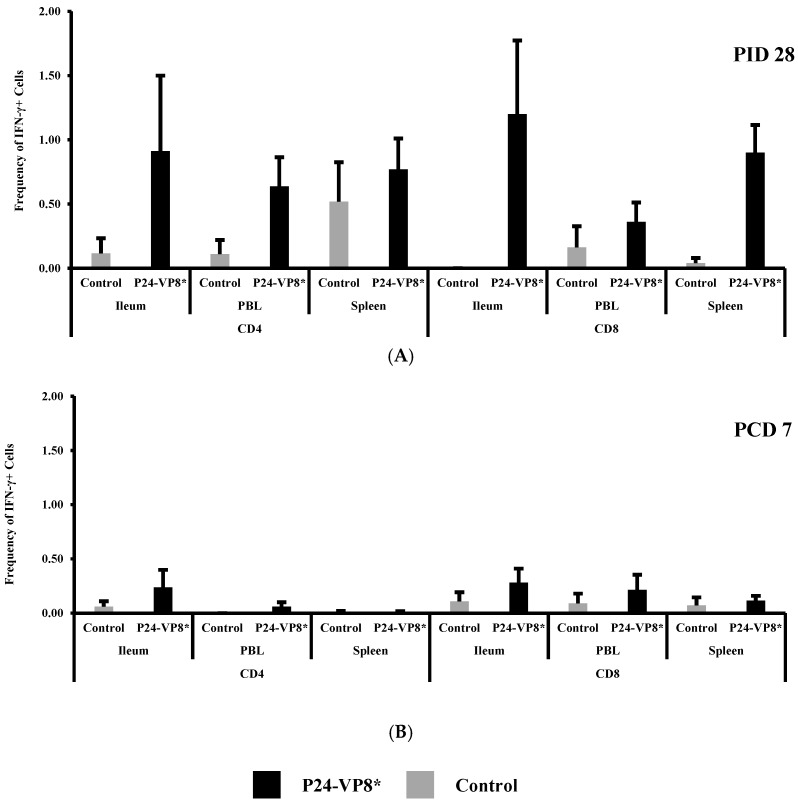
Frequencies of IFN-γ+CD8+ and IFN-γ+CD4+ T cells in ileum, peripheral blood (PBL), and spleen at PID 28 (**A**) and PCD 7 (**B**). Two-way ANOVA followed by Multiple *t*-tests were carried out for comparisons. (*n* = 3–8; *p* < 0.05). There were no significant differences.

**Table 1 vaccines-07-00177-t001:** Assignment of treatment groups for gnotobiotic (Gn) pigs.

Group	Number of Pigs	Challenge	Time of Euthanasia
Control	3	No	PID 28
Control	7	Yes	PCD 7
P24-VP8* Vaccine	7	No	PID 28
P24-VP8* Vaccine	8	Yes	PCD 7

**Table 2 vaccines-07-00177-t002:** Diarrhea and rotavirus fecal virus shedding in Gn pigs after the VirHRV challenge.

		Diarrhea				Fecal Virus Shedding				
Treatment	*n*	% with Diarrhea ^a^	Mean Days to Onset ^b^	Mean Duration Days ^c,§^	Mean Cumulative Fecal Score ^c,^*	% Shedding Virus ^a^	Mean Days to Onset ^b^	Mean Duration Days ^c,^*	Mean Peak Titer (FFU/mL)	AUC
P24-VP8*	8	87.5	4.4 (0.5) ^d,^*	3.3 (0.75) *	9.1 (1.23) *	75	4.8 (1.0)	2.5 (0.89) *	8500 (2196) *	11,750 (3172)
Control	7	100	1.6 (0.3)	6.0 (0)	14.3 (0.44)	85.7	1.9 (0.14)	5.9 (0.14)	11,492 (4300)	26,664 (10,489)

^a^ Gn pigs were orally inoculated with 1 × 10^5^ FFU/mL of VirHRV at post-innoculation day (PID) 28. Rectal swabs were collected daily after the challenge from PCD 1 to PCD 7 to monitor for clinical signs and virus shedding. Pigs with daily fecal scores of ≥2 were considered diarrheic. Fecal consistency was scored as follows: 0, normal; 1, pasty; 2, semi-liquid; and 3, liquid. Fecal virus shedding data was determined by ELISA and/or CCIF. ^b^ An arbitrary designation of Day 8 was assigned to pigs that did not develop diarrhea or shed virus in feces for calculating the mean days to onset. ^c^ For the purposes of calculating diarrhea and virus shedding duration, if no diarrhea or virus shedding was observed in pigs until euthanasia day (PCD 7), the duration days were recorded as 0. ^d^ Standard error of the mean. ^§^ Student’s *t*-test was used for comparison between vaccine and control groups. Asterisk indicates statistical significance between the groups (*n* = 7–8; *, *p* < 0.05).

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
