# Peer review of "Parenterally Administered P24-VP8* Nanoparticle Vaccine Conferred Strong Protection against Rotavirus Diarrhea and Virus Shedding in Gnotobiotic Pigs"

_vaccines, 2019, doi:10.3390/vaccines7040177_

Round 1

Reviewer 1 Report

This is a well-written report on the immunogenicity and protective efficacy of novel nanoparticle vaccine against rotavirus diarrhea in gnotobiotic pigs. It also well described the limitations of the study. In the further study, interesting results are expected through various rotavirus and norovirus experiments using gnotobiotic pigs.

minor comment

How was the dose of the P24-VP8 vaccine determined?

Reviewer 2 Report

Ramesh et al have evaluated the suitability of P24-VP8* nanoparticle vaccine to prevent human rotavirus infections.

One question regarding the title is the word “strong” as the results clearly show the incidence of diarrhea and virus shedding has not been significantly reduced after the administration of vaccine.

Figure 1(b), after 5 days post-challenge the FFU values are the same even though the diarrhea scores are different in 1(a). What can this data tell about the vaccine?

Methodology related to Gn pigs experiment can be explained better.

Reviewer 3 Report

A very well written and conducted study of an important health issue. These findings are applicable not only to the human health condition but as well as an approach that can be applied to animal health.  If the efficacy is proven under extensive field testing it may result in a more cost effective and therefore available immunization program. This approch should also avoid the rare but serious problem of intussusception with oral vaccines. A well done study of interest to a significant audience.

Author Response

Thank you!

Reviewer 4 Report

The article by Ashwin Ramesh et al. demonstrates on the gnotobiotic pigs the effect of a P24-V8* rotavirus vaccine.

I recommend publication after major modifications according to the following comments and advice.

Major comments :

Could the authors give more detail about the number of passages for paragraph 2.1 (27th passage for VirHRV and 35th for AttHRV)? could you justify the absence of confusing bias regarding the different numbers of passage for these two viruses?

The vaccine (paragraph 2.2) was indicated as stored up to 8 months. Could the authors justify the stability of their vaccines? Could they stratify the observed results regarding the delay of conservation before administration to pigs?

Could the authors justify the administration plan (1 injection and 2 boosts)? Have they experimented with other timing or number of injections?

Could the author justify the delay before euthanasia to 7 days? Have they verified if a supplementary delay was associated with other effects (on PCD7+ outcomes and measures)?

Why do the authors do not measure the viral shredding using qPCR?

Figure 2: have the authors tested negative serums without dilution to ensure the absence of Ig? If think this experiment has to be done to ensure robust comparison.

Figure 3: It is very surprising that no significant difference could be observed (especially for CD4+ in the ileum or PBL or CD8 in Ileum or spleen at PID28 and Ileum CD4+ at PCD7). Could the authors give more detail or modify the figure to represent IC95?

Minor comments : 

The authors have to detail all the acronyms used in the manuscript at their first use (example: GAVI - line 40)

Please detail and give reference fo the sentence line 45-46 "they are also not [...] oral vaccines in LMICs"

A paragraph at the end of the introduction giving detail about the precise objective of the study has to be added.

The authors administered NaHCO3 before vaccine administration. How do they imagine to apply this to humans' vaccination?

How the authors justify the use of this scale for fecal consistency and not a validated one (as Bristol's scale)?

Could the authors justify the choice of cell stimulation using 12ug/ml of P24-VP8*?

Figure 1: diarrhea was considered for a score > or =2 or >2 only?

Figure 2: Identification of comparison with or without significant difference is not easy to understand with the chosen methodology. Please simplify.

Line 229 : data not shown : please give more details or give the information in suppl. data.

Round 2

Reviewer 4 Report

The authors improve significantly the manuscript as adviced by the reviewers. 

I recommend publication of this very interesting manuscript after adding all these clear explanations in the manuscript to ease the understanding for non-specialist of Gn pigs or Rotavirus. 

Author Response

Thanks for your suggestion. We have added the relevant contents to the manuscript to make it easier for readers who are not Gn pig and rotavirus specialist.